# Experiences of New Zealand Haemodialysis Patients in Relation to Food and Nutrition Management: A Qualitative Study

**DOI:** 10.3390/nu13072299

**Published:** 2021-07-03

**Authors:** Rachael M. McLean, Zhengxiu Xie, Vicky Nelson, Vili Nosa, Hla Thein, Audrey Po’e-Tofaeono, Robert Walker, Emma H. Wyeth

**Affiliations:** 1Department of Preventive and Social Medicine, Dunedin School of Medicine, University of Otago, 18 Frederick St, Dunedin 9016, New Zealand; zhengxiu.xie@otago.ac.nz; 2Ngāi Tahu Māori Health Research Unit, Dunedin School of Medicine, University of Otago, 18 Frederick St, Dunedin 9016, New Zealand; vicky.nelson@otago.ac.nz (V.N.); emma.wyeth@otago.ac.nz (E.H.W.); 3Faculty of Medical and Health Sciences, University of Auckland, 28 Park Ave, Grafton, Auckland 1023, New Zealand; v.nosa@auckland.ac.nz (V.N.); apoe372@aucklanduni.ac.nz (A.P.-T.); 4Counties Manukau District Health Board, 100 Hospital Road, Otahuhu, Auckland 1640, New Zealand; Hla.Thein@middlemore.co.nz; 5Department of Medicine, Dunedin School of Medicine, University of Otago, P.O. Box 56, Dunedin 9016, New Zealand; rob.walker@otago.ac.nz

**Keywords:** kidney disease, haemodialysis, New Zealand, qualitative research, diet

## Abstract

People receiving haemodialysis have considerable and complex dietary and healthcare needs, including co-morbidities. A recent New Zealand study has shown that few patients on haemodialysis are able to meet nutritional requirements for haemodialysis. This study aims to describe the perspectives and experiences of dietary management among patients on haemodialysis in New Zealand. This exploratory qualitative study used in-depth semi-structured interviews. Purposive sampling was used to recruit participants from different ethnic groups. Forty interviews were conducted, audio-recorded and transcribed verbatim. An inductive approach was taken using thematic analysis. Forty participants were interviewed. Participants spoke of major disruption to their lives as a result of their chronic kidney disease and being on haemodialysis, including loss of employment, financial challenges, loss of independence, social isolation and increased reliance on extended family. Most had received adequate dietary information, although some felt that more culturally appropriate support would have enabled a healthier diet. These findings show that further support to make the recommended dietary changes while on haemodialysis should focus on socio-cultural factors, in addition to the information already provided.

## 1. Introduction

Chronic kidney disease (CKD) is a leading cause of morbidity and mortality globally [1]. In New Zealand, CKD accounts for around 1% of total disability-adjusted life years [2], due to resulting premature mortality and disability. In New Zealand, in 2019, there were nearly 3000 people living on dialysis (around 70% of these on haemodialysis) and 656 new patients requiring renal replacement therapy [3].

There are substantial ethnic inequities in both the incidence and prevalence of end stage kidney disease in New Zealand. The incidence of renal replacement therapy is substantially higher for Māori (indigenous New Zealanders), Pacific and Asian people, than Europeans, proportional to population [3]. Māori are more likely to commence renal replacement therapy with haemodialysis and less likely to receive a renal transplant than non-Māori. Māori and Pacific people are more likely to experience CKD at an earlier age than non-Māori, non-Pacific people [3], have higher prevalence of co-morbidities, such as diabetes, and are more likely to live in more deprived neighborhoods than Europeans [4].

People receiving haemodialysis have considerable and complex dietary and healthcare needs, including specific dietary and fluid intake requirements that differ from those of the general population [5]. Patients may also have to juggle the complex needs of dialysis treatment with those of other co-morbidities, such as diabetes and cardiovascular disease. Restrictions on intakes of fluid, sodium, potassium, phosphorus and saturated fat, as well as modification of protein intake, can be burdensome and difficult to manage. A recent New Zealand study undertaken by our team has shown that few patients on haemodialysis are able to meet all of these nutritional requirements [6]. Evidence shows that optimal nutrition while on haemodialysis results in better quality of life, reduced symptoms and fewer complications [5,7]. Therefore, it is important to understand barriers for meeting the nutritional needs in order to support patients on haemodialysis and develop interventions to improve nutrition and quality of life in this group.

This study aims to describe the perspectives and experiences of dietary management among patients on haemodialysis in New Zealand. We sought to identify the barriers to and enablers of following recommended dietary requirements and learn more about specific needs to allow the development of culturally appropriate nutrition resources and support.

## 2. Materials and Methods

### 2.1. Study Design

This exploratory qualitative study used in-depth semi-structured interviews. The standards for reporting qualitative research (SRQR) [8] have been followed for reporting. Ethical approval was obtained from the University of Otago Human Ethics Committee (Health, H19/050). Written informed consent was obtained from all participants involved in the study.

Our multidisciplinary research team consisted of both clinicians involved in dialysis treatment and independent researchers, including a dietitian, and researchers experienced in mixed methods research. The clinicians’ role was to facilitate appropriate access to participants, advise on settings for interviews, and contribute to the interview schedule. Clinicians were not involved in interviewing participants to minimize social desirability bias of responses and bias of interviewers. To maximize communication, enhance understanding and interpretation between researcher and participants, we had an ethnically diverse team of researchers.

### 2.2. Participant Recruitment

Purposive sampling [9] was used to recruit participants from different ethnic groups in New Zealand: European, Māori, Pacific and Asian [10]. Participants were adults (18 years of age or older) who were currently receiving haemodialysis treatment for end stage kidney disease, who were living either in the Southern South Island region or Auckland (New Zealand’s largest and most ethnically diverse) city. Ethnicity was collected using the standard NZ Census ethnicity question [10], where participants were able to identify multiple ethnic groups if they wished. The majority of the interviews were conducted in English. Four interviews were conducted in Mandarin and one was conducted in Cantonese with an interpreter. Participants were recruited through dialysis units by the researchers in those areas. Recruitment ended when data saturation was reached.

### 2.3. Data Collection

Participants took part in semi-structured interviews [9]. The interview schedule was initially developed by RM and ZX, based on one used previously in our research group, as well as a review of relevant literature. All members of the multidisciplinary research team contributed to revisions of the interview schedule.

Interviews were conducted either in the participant’s home or in the dialysis unit, depending on participant’s preference and convenience. Māori and European participants were interviewed by a researcher from the same ethnic group. A Chinese researcher interviewed Asian participants and a Tongan researcher interviewed all Pacific participants. Participants were initially asked about their general experience of being on dialysis and coping with dialysis. The main focus was on their experiences relating to managing food and nutrition to identify the barriers/enablers, such as support from health providers, family/friends/community groups, how easy to follow the dietary advice, the adequacy of the information they received and their understanding around dietary advice (Appendix A). The interviews were audio-recorded, with consent. The interview audio recordings were transcribed verbatim using a transcription service and checked for accuracy by the researchers.

### 2.4. Data Analysis

All transcripts were reviewed by RM and ZX. An inductive approach was taken using thematic analysis [11,12]. Data were grouped in domains (from the interview schedule) and themes were generated by RM and reviewed with ZX. Investigator triangulation was undertaken to finalize themes and maximize the input and experiences of our multidisciplinary team. V. Nelson, EW, AT and V. Nosa contributed to the generation of themes for Māori and Pacific interviews and all authors contributed to final interpretation. Coding was completed with the aid of the NVivo software [13]. We sought to identify and develop themes based on common experiences, as well as exploring the wide range of experiences, with a focus on reporting experiences of people across ethnic groups. Representative and reflective quotes were identified and annotated according to ethnic groups, Māori (M), Pacific (P), Asian (A) and European (E), and the participant number within the ethnicity grouping (e.g., M3 refers to Māori participant 3). To preserve anonymity, the prioritized ethnicity method has been used where Pacific and Asian are aggregates of multiple ethnicities [10].

## 3. Results

Forty participants were interviewed; two were interviewed by telephone (one by preference and one because major flooding had closed the roads) and 38 in person. Of these, four were interviewed in their homes (because they dialyzed at home) and 34 were interviewed in the dialysis unit in Counties Manukau or Dunedin Hospitals. Demographic information is presented in Table 1; 42.5% of participants were female (n = 17), 5 participants identified as Māori and eight as European, 15 participants identified with Pacific (including Samoan, Tongan, Niuean, Fijian and Cook Island Māori) and 12 with Asian (including Chinese, Indian, Sri Lankan) ethnic groups. Age of participants ranged from 26 to 89 years with a mean age of 63 years.

### 3.1. Major Disruption

Participants spoke of major disruption to their lives as a result of their chronic kidney disease and being on haemodialysis: “it’s difficult to adjust. And it doesn’t sort of fit with my lifestyle and my home” (P5). Some participants reported being overwhelmed by the large amount of information they had to absorb at the time of commencing dialysis, which included information about diet, and the large amount of written material that accompanied this information. At the same time, a number of participants spoke of coming to terms with going on dialysis, the changes this entailed, and grieving for their life before starting dialysis. Several Māori participants spoke about going through an initial denial phase, particularly at the time of receiving a large amount of information: “I just didn’t want to listen. I didn’t want to admit that I was gonna be a dialysis patient” (M4). Although the majority of participants agreed that dietary change was important for their health, many found this challenging due to the multitude of other disruptions in their lives. Almost all participants had made changes to their diet at some stage and monitored fluid intake, although the degree to which they were able or willing to do this varied considerably. The major disruption associated with starting dialysis meant that some, particularly those with limited financial resources, found it difficult to adhere to dietary guidelines, while others were unable to prioritize diet above other competing demands.

#### 3.1.1. Social Isolation

Starting dialysis meant that many participants had to give up paid employment with resulting financial challenges, loss of independence, and social isolation: “I’m nearly 40 years old now, this was the time when I would have been working for something and building for my family or my career...but because of dialysis, it all stopped” (A10). Others had significant changes in their living situation such as having to move in with relatives for support, having to move to a different city to be closer to medical care, or moving into rest home care. Some reported being unable to travel domestically or internationally due to dialysis commitments, which meant that they could no longer visit close family members: “there are many problems I always see, like, I’m unable to go anywhere, like overseas, Tonga and America. Because in Tonga, there is no dialysis” (P7). For some this resulted in increased social isolation and disruption of family relationships: “I miss my friends a lot. Never see them again” (M1). There was also difficulty in explaining to friends and whānau (family), why they could not commit to social functions or events due to dialysis: “people don’t understand, like friends texting me going ‘What are you up to?’ and I go, ‘I’m at dialysis, remember, Monday, Wednesday, Friday’” (M2).

#### 3.1.2. Changing Relationships with Family/Whānau

For many participants, the major disruption associated with being on haemodialysis was related to important changes to their relationships with family/whānau. Most became more dependent on partners and/or family members for care and support, including for the provision of food. A number of Māori participants mentioned trying to relay information about their diet and its importance to whānau, as they lived with whānau who were responsible for the shopping and cooking: “even my family were involved with what was happening…so the whānau. It was really trying to get them to understand what I was going through and ‘cause everybody was a bit confused” (M3). For some who were already in living situations where their partner was the primary food provider in the household (purchaser and cook), the change involved working with their partner to learn and adapt to the new dietary regime. Others spoke of not wanting to burden their families with complex dietary requirements: “it’s hard for them because when they cook, they have to cook it, cook my food first...but then take my food out because I don’t like salt. They find it hard” (P6). Some kept information regarding dietary needs from family members for this reason. Those who lived with extended family faced particular challenges with balancing their own dietary needs, not wanting to burden family members and being grateful for the food that was provided, however unsuitable. This created a sense that whānau did not care about their health and diet needs: “they don’t care. They think oh he’s sick, I’m not, so I’m gonna eat this” (M3). Some chose to avoid family meals, rather than offend family members by identifying that the food provided was not suitable for them.

#### 3.1.3. Dependence and Loss of Control

Being on dialysis meant that most participants became dependent on others for support and for many this meant a loss of control over the food they were able to consume. While many were able to work with their spouse to ensure that dietary changes were made in line with recommendations, others were unable to influence the food that was available. This was particularly the case for those who had to move in with extended family, and those in elder care homes: “they [rest home] provide food for me, but, it’s not to my liking…they don’t give me meat…they don’t allow us to cook (A9)”. Some reported that they were grateful that their relatives were able to care for them, as they themselves were unable to cook due to physical impairments and/or financial limitations. Often family members were in charge of both the cooking and food shopping. As a result, they did not feel empowered to direct they type of food that was offered: “you know how Pacific Islanders cook? Just for the whole family, so I have to adjust myself and eat what is given to me. Otherwise, I can’t survive” (P3). This was an experience shared by a number of Māori participants who lived with whānau. They found it difficult to communicate the necessary changes in their diet, particularly when whānau took care of the shopping and cooking: “they do my shopping” (M5). This was due to whānau members finding the prospect of changes and illness ‘scary’: “they don’t like hearing it…scary (M5)”. Others, with limited or no English language, were dependent on family to communicate with healthcare professionals and translate messages. One was grateful to see a dietitian who spoke Chinese so that they could understand what dietary changes were recommended: “because he [dietitian] comes from China and can talk to me in Chinese. Make sense in the language I understand” (A8).

### 3.2. Independence, Adherence and Control

Several factors were associated with independence and feelings of autonomy and ongoing attention to maximizing dietary change. These included family/whānau support (often a supportive partner), financial security, ability to maintain paid employment, and ability to dialyze at home. Those who were financially secure (either through being able to continue working, or because of other financial resources) found it easier to move into more comfortable and suitable accommodation close to healthcare providers. They were also more able to seek out palatable and appropriate foods than those on low incomes: “and protein, obviously, is the salmon, steak, uh, any of that high-end fish, blue cod. I look at the top end of the market there.... I have a wonderful diet” (E8). Many participants had to stop working and became dependent on government assistance in the form of benefits. Several reported that government funding was inadequate and that they needed more financial support in order to eat properly.

#### 3.2.1. Constant Monitoring

A sense of control was demonstrated by participants constantly monitoring their blood results, weight and blood pressure and aligning these with food and fluid requirements. Others found being on dialysis overwhelming and were unable to do this on top of other competing demands.

#### 3.2.2. Attitude

Many participants also reported attitudinal approaches to dialysis and eating that enhanced maintenance of a sense of control: “I live life to the full. And you know, there will probably come a day when I can’t. But I’m hoping to get to 92” (E7). Other participants talked about how they could ‘get away’ with eating certain foods, such as fast food, because it wasn’t on the list of foods to avoid whilst on dialysis: “burgers and all that sort of stuff. It’s not actually the stuff that is on that list. It’s all the carbs and all that but the carbs and that aren’t on those lists, they are the electrolytes you’re worried about” (M2). One participant also spoke about enjoying more foods that should be avoided on dialysis day, “because dialysis is just gonna take it out”, “like today [dialysis day] I put tomato slices in my sandwich and you’re not meant to” (M2).

#### 3.2.3. Peer Support

Several participants valued the ability to connect with others also on dialysis to share knowledge and provide support through organizations such as the Kidney Society or Dialysis Society. Others wished that they had more opportunity to connect with dialysis patients, online, through a regular newsletter, or in person. Some participants mentioned having more peer support would be helpful, particularly at the beginning of dialysis and understanding the process, social changes, ‘grieving process’ for their old life, physical changes they may go through. This would help in connecting with others that understood the process, as they had been through it before and may have tips or advice on some of the symptoms or changes to expect and how to deal with them: “’cause they understood what I was going through. I didn’t. They understood that, you know, this is something that actually happens to people often and quite, are normal. It’s a normal process… If you had like a, I dunno, a get-together or a support group or something to ask people” (M3).

### 3.3. Importance of Appropriate Professional Support

Support from family and healthcare workers was vital for all participants.

#### 3.3.1. Information—Both Verbal and Written

Most participants reported that they had received adequate information about managing their diet while on dialysis. They reported having seen a dietitian for individualized support and advice at least once at the start of dialysis and some had ongoing support from the dietetic service. Others had had dietary advice from other health professionals including renal physicians and nursing staff. For most there was no formal follow up with participants about dietary practices. Diet was only discussed when laboratory results were abnormal (such as if serum potassium was elevated), or when acute adverse events occurred. However most participants felt that if they had questions they could access further information as required: “uh, I spoke to a dietitian … and I don’t think I’ve had contact with her since. She gave me a diet sheet and all that sort of thing. So I didn’t feel the need to speak to her again. If I had, uh, called and, said to the staff, ‘I needed- I’d like to talk to a dietitian’, I-I would get her straightaway” (NZE3).

All participants recalled being given generic written information about what dietary changes are necessary while on dialysis. For most, this information was provided at the time of starting dialysis and was given to them by either a nurse or dietitian in the dialysis unit. For a minority, especially those who started on dialysis following what appeared to be a short illness period, they were only made aware of the importance of dietary changes following complications of their dialysis treatment: “I never got told anything about the food or what to eat or anything. Never. Until this year in March, I ended up in hospital with high potassium. ‘Cause I was eating everything and anything and I would just cook mashed potatoes to eat…” (M2). Participants found written information useful and many kept it to hand in the kitchen or while shopping. Paper sheets with pictures and simple instructions were particularly valued, in preference to electronic resources: “I’ve got them on my fridge and I’ve got pictures, when I’m out I’ll look at my photos. The charts, yeh, from the nutritionist. And I just follow those, I don’t, but when we shop, I shop to those color factors, no longer buy silver beet” (M2). Information from health professionals was trusted and very few participants reported using online resources for dietary information.

Some, particularly Pacific participants, felt that the information they received was not culturally appropriate and this was a barrier to them being able to eat healthy, culturally appropriate food: “so even like, taro leaves is no good for us. I can’t remember why. I don’t know if it’s the fat in the leaves or the starch. I just can’t remember. But they said you can’t eat too much of that” (P3). Provision of verbal information in different languages would also have benefited understanding: “a lot of older Pacific people that are on dialysis. A lot of them hardly speak English or English is their second language, so I think we need somebody that can help them understand” (P8). Others found that having health professionals from similar ethnic backgrounds enhanced communication and understanding and provided the catalyst for change: “yeah, because before I don’t eat the way they tell me to eat. Now I follow. The Tongan doctor came and said do you want to live to see your grandkids? Then after that, he changed his mind. He really helped him to follow the advice” (P11).

#### 3.3.2. Dietary Information and Changes after Medical Complications

Some participants reported adhering to the required dietary changes more closely only after they had encountered a major complication of dialysis. These participants reported only receiving the appropriate and sufficient information on the required diet changes after such events. One participant described getting information on their diet and the need to monitor serum potassium levels, after going into heart failure: “...then they do dialysis a few times to sort of take all the potassium out and then I was like, They said, ‘Oh what have you been eating?’ I’m like, ‘What do you mean?’ And then I got the sheets, the um…the traffic lights. And I live by that now…It was like they missed giving me all the information” (M2). Others reported following dietary advice becoming important after experiencing a major medical complication: “it’s, it wasn’t really important to me in the beginning but as time has gone on, it’s becoming, like the hospital’s really helped me understand that I need to be careful of things” (M3). One participant who did not follow dietary advice closely, said that they would be more careful about their diet if they did become unwell: “I know there is times where I will change a bit of my diet.” “I’m adamant not, um for that not to happen. Especially I worry about hand and my feet, gangrene. That’s all I think about” (M4).

#### 3.3.3. Relationships with Health Providers

All participants talked of the importance of having ongoing relationships with trusted health providers. For most, this involved regular face to face meetings and consultations, supported by their availability over the telephone for additional questions or concerns. Having ongoing ready access to information and advice was important and all participants felt that this was available to them. The importance of having a collaborative multi-disciplinary team was highlighted with participants valuing support from dietitians, social workers, administrative staff, and nursing and medical professionals. This was particularly noted by those that dialyzed within clinics, such as the ‘Super Clinic’ in Auckland: “like there’s probably 30 nurses and you get to know all their names after a few, every time we’ll have a different nurse each time and they do all their and this is an awesome unit that’s helped heaps. ‘Cause they’re also full of knowledge” (M2). Only one participant reported looking for dietary information from websites and other information sources. The majority trusted and relied on information provided by their healthcare team. 

### 3.4. Dietary Changes and Challenges

Dietary changes reported by participants included limiting fluid and salt intake and being mindful of potassium intake by limiting intake of certain fruit and vegetables. While some reported that they had never been big salt consumers, others found it difficult to reduce salt and some reported using alternatives, such as pepper and spice mixes. Nearly all participants reported that limiting fluid intake was particularly challenging. Several participants reported missing favorite foods, such as seasonal fruits: “and, I have what I call a naughty season and we’re into it now. It’s fruit. … Nearly every fruit is banned. I’m allowed to eat lemons (laughs)” (E8). A few participants commented on the challenge of accessing convenient healthy foods in an environment that promotes availability of unhealthy options.

#### 3.4.1. Maintaining Cultural Identity and Dietary Practices

Many participants talked of the importance of being able to continue eating food that they were used to eating, especially traditional cultural food. This enhanced their enjoyment of food, maintained their cultural identity, and allowed them to continue to participate in family, community, and cultural events. However, the degree to which participants were able to adhere to the diets they wanted varied according to financial resources and reliance on others: “when you are in the hospital, you need to strictly follow what they say...European meals were provided, but we Chinese are not familiar with it. I mainly eat Chinese meal” (A4). Māori participants reported not feeling as though their culture was a part of their treatment or dietary changes. This was due to a number of reasons, including the inability to eat traditional foods due to the dietary changes required. Further, some participants described being seen as a ‘dialysis patient’ first rather than Māori, making them feel as if their dialysis status came before their cultural background and practices: “I just think they take into account dialysis and then you have to sort of work it around it yourself” (M2).

#### 3.4.2. Physical Symptoms

Physical symptoms impacted on participants’ ability to eat at times. Many reported changes in taste, lack of appetite or variable appetite on dialysis and non-dialysis days. Food that previously they had enjoyed became bland, tasteless and unpalatable. For others, some foods made them feel nauseated and several participants reported that some foods made them vomit (e.g., meat). Other symptoms included tiredness and lethargy, poor vision and mobility, which made shopping and cooking difficult. Fatigue and lethargy sometimes led participants to look for easy meals, such as takeaways, even though they knew that these were not healthy options.

#### 3.4.3. Juggling Complex Dietary Needs of Co-Morbidities

Many participants had other chronic conditions in addition to their CKD. Many of these conditions also required dietary modification: “it’s lots of things you learn about those conditions. What to eat and when your blood sugar is low, the blood pressure, you about how to adjust. It’s just the knowledge. You have to have the knowledge of the food” (P11). For some, poor oral health affected ability to eat recommended foods: “I’ve got no teeth [laughs] on the top” (M5). Participants highlighted the complication of taking medication for other conditions and ‘juggling’ this with dialysis and these medications, either requiring to be taken with food, or without: “well when they start that early you don’t take your meds for a start ‘cause some of them interfere with the session. So there’s breakfast gone ‘cause some of them you gotta take while you eat” (M1). Further highlighted was the complication of medications interfering with dialysis and vice versa: “it’s just certain things and certain items that will clash with what you’re taking every day like the diabetes or dialysis” (M4).

## 4. Discussion

Being on haemodialysis resulted in major disruption to peoples’ lives involving changes in living and working situations, profound loss of control, changing relationships with family/whānau and reliance on ongoing support from health professionals. The majority of people in this study reported that they believed dietary change was important for their health and that they had been provided enough information by health professionals involved in their care, including dietitians. Their ability to make the necessary changes varied according to their financial security, living situation, and sense of control over their lives. Those who were reliant on wider family for providing food found it particularly difficult to control their diet, but were grateful for the support they were getting. Complications of end stage CKD, such as fatigue, altered taste perception, poor oral health and tooth loss, affected food intake. Patients also reported limited time for shopping/food preparation, trouble selecting the right food and difficulty integrating renal dietary advice into family meals.

All participants valued the care and support they received from trusted health professionals and spoke of the importance of long-term professional relationships with healthcare providers. Having written and pictorial information, supported by ongoing and on-call advice and support, was particularly valuable. Few participants in this study sought dietary information from outside sources, such as websites or social media.

In general, lack of information was not a barrier to maintaining adherence to dietary guidelines in this study. Barriers were mainly socio-cultural and included lack of money (for participants and their family/whānau) and lack of independence and control over living situation, including diet. Involvement of wider family/whānau in in-depth discussions about not only what dietary changes are needed, but also how these can be achieved within a patient’s living situation and cultural context are also needed.

Food insecurity (defined as an “[in]ability to acquire nutritionally adequate and safe food that meets cultural needs and has been acquired in a socially acceptable way” [14]) has been described in New Zealand over many years and is socially patterned with Māori, Pacific and those on low incomes over-represented in food insecure households [14,15]. Food insecurity is associated with poor health outcomes, including psychological distress and poor diet quality [14,16,17]. Efforts to improve the diet quality of those on haemodialysis need to focus on wider food security issues in families and communities, particularly for those with low incomes, or those who find themselves having to rely on others as a result of their illness. Although this was not a representative sample of New Zealand dialysis patients, we deliberately recruited participants from groups that experience inequities in health outcomes by ethnicity. Māori, Pacific and Asian ethnic groups are over-represented among those on dialysis [3] and Māori and Pacific people are often diagnosed later with end stage renal failure and therefore dialysis treatment may be delayed [18,19]. In this study, Māori, Pacific and Asian participants in particular often reported financial difficulties, loss of employment, and loss of independence. Some participants lived with extended family for better support, which was mostly positive, but presented some challenges. Other challenges included those of managing co-morbidities and meal planning around dialysis timing.

Maintaining access to foods that are culturally familiar or important was difficult for many participants, which exacerbated feelings of loss of control and dislocation from normal life. Participants reported that, in order to follow the food guidelines they had been given, they had to give up food that they loved, such as tītī (muttonbirds, a traditional Māori delicacy), Fijian and Chinese food. Others reported that the information provided was not tailored for Pacific dietary practices. More opportunities for peer support may enable patients and their families to devise and share solutions amongst others in similar situations. Pacific and Asian participants also reported a sense of loss at being unable to travel to and connect with family overseas. Given the relative over-representation of Māori, Pacific and Asian people on haemodialysis, these findings represent a potential source of inequities of outcomes in terms of risk of medical complications, as well as decreased quality of life.

Recent qualitative research among Māori with CKD has shown that many experienced marginalization in the healthcare system and felt the cultural considerations were often not considered [19]. Another study which focused on developing culturally appropriate nutrition resources using focus groups with healthy Māori and Pacific people showed the importance of using information that was culturally relevant, including language and traditional foods and practices [20]. Food cost was also identified as a barrier to healthy eating, especially for those with large families [20].

International studies have also shown a range of factors may be involved, including disease and socio-cultural factors [21]. A recent Australian study among HD patients showed that competing medical and social issues coupled with contradictory and/or irrelevant dietary advice were barriers to the implementation of dietary and fluid restrictions [22].

This study includes a cross-section of New Zealand dialysis patients, including those from urban and rural settings and different age and ethnic groups. Participants were recruited from Auckland and the Southern region, which covers a large geographic and rural area. Although this is not a representative sample of those on haemodialysis in New Zealand, we believe we have been able to capture a wide range of experiences, including from participants from ethnic groups who experience poorer health outcomes and are over-represented in the dialysis population. Māori, Pacific, Chinese and European researchers conducted and interpreted interviews with participants of their own ethnic group in order to enhance understanding and insights. There are some limitations. The majority of interviews were conducted in English, which may have limited participation for some participants. Although we have a wide range of participants in this study, there may be opinions and experiences that were not captured that would have added to our findings.

## 5. Conclusions

These findings show that further support to make the recommended dietary changes while on haemodialysis should focus on socio-cultural factors. Although provision of adequate information is essential, lack of information was not a major barrier for most people who struggled to meet recommendations. Instead, financial security was an important enabler and the provision of culturally appropriate support was essential for this group. Partnerships between healthcare providers and community leaders may also maximize cultural communication. Practical support for those dependent on extended family may also enhance resilience and dietary adherence.

## Figures and Tables

**Table 1 nutrients-13-02299-t001:** Participant characteristics.

Demographic Characteristics	Number (%)
Gender	
Female	17 (42.5)
Male	23 (57.5)
Age	
Mean(standard deviation)	63 (15)
Range	26–89
Ethnic group	
Māori	5 (12.5)
Pacific	15 (37.5)
Asian	12 (30.0)
European	8 (20.0)
Interview setting	
Dialysis unit	33 (82.5)
Home	5 (12.5)
Telephone	2 (5.0)
Total	40

## Data Availability

Data described in the manuscript will be made available upon reasonable request from the corresponding author.

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
