# Peer review of "Experiences of New Zealand Haemodialysis Patients in Relation to Food and Nutrition Management: A Qualitative Study"

_nutrients, 2021, doi:10.3390/nu13072299_

Round 1

Reviewer 1 Report

Rachael Mira McLean and colleagues realize, in an interesting way, an interview-based report on people receiving chronic haemodialysis, which describes the perspectives and experiences of dietary management among forty New Zealander patients.

The major objective of the report was to describe haemodialysis patients’ attitude when asked to deal with complex diet in addition to healthcare needs, including co-morbidities. The majority of patients, when undergoing on haemodialysis, had extreme difficulties to adapt to new dietary changes required from the disease.

This is a good article, but some minor revisions are need:

  • I suggest to delay some off-topics point: as the article is based on haemodialysis patients’ food and nutrition management, paragraph regarding other kind of challenges these patients have to deal with, once starting haemodialysis, are irrelevant.
  • I suggest to improve English grammar: there are some mistakes which, in some cases, make difficult to understand what the authors want to say (for example, “ we sought to identify the barriers and enablers to following recommended dietary requirements…”/ “ We sought to identify and develop themes based on common experiences, as well as reflecting the wide range of experiences..” ). Please specify. 

The only table in the article is clear.

Author Response

We would like to thank the two anonymous reviewers for their support of our research, and their useful feedback.  We have responded to Reviewer 1’s suggestions as below.

Reviewer 1

Rachael Mira McLean and colleagues realize, in an interesting way, an interview-based report on people receiving chronic haemodialysis, which describes the perspectives and experiences of dietary management among forty New Zealander patients.

The major objective of the report was to describe haemodialysis patients’ attitude when asked to deal with complex diet in addition to healthcare needs, including co-morbidities. The majority of patients, when undergoing on haemodialysis, had extreme difficulties to adapt to new dietary changes required from the disease.

This is a good article, but some minor revisions are need:

  • I suggest to delay some off-topics point: as the article is based on haemodialysis patients’ food and nutrition management, paragraph regarding other kind of challenges these patients have to deal with, once starting haemodialysis, are irrelevant. 

Thank you for this suggestion.  We have added some more information to section 3.3.1 which relates particularly to dietary information.  However we believe that the other very important challenges faced by participants are directly relevant to why patients are unable to prioritise their dietary changes, and have not removed these other sections.

  • I suggest to improve English grammar: there are some mistakes which, in some cases, make difficult to understand what the authors want to say (for example, “ we sought to identify the barriers and enablers to following recommended dietary requirements…”/ “ We sought to identify and develop themes based on common experiences, as well as reflecting the wide range of experiences..” ). Please specify. 

Thank you, we have revised these phrases in the text.

The only table in the article is clear.

Reviewer 2 Report

Thank you for undertaking this study to improve understanding of the barriers and enablers those who require haemodialysis as a life sustaining treatment experience relating to food and nutrition. The patient experience and perspective is clearly described and highlights a number of areas where the delivery of nutrition information can be improved. However there is inadequate information in the manuscript about the setting and how dietary information is provided, and methodological considerations, particularly around data collection and the consideration of bias and assumptions during the inductive process of deriving themes. The manuscript would be improved if more context could be provided around how and when dietary information is provided to this group. For example, is dietary information tailored to individuals, or is generic information provided to everyone? How often are patients able to access dietitians to ask questions? Are patients provided with feedback regularly about positive and negative dietary compliance, or are they only approached when potentially non-compliant? How were the interview questions derived and who was involved? What was the impetus for the research? There is inadequate information about the research team included in the manuscript. What relationship exists between the researchers and participants? What bias, assumptions and background do the researchers identify about themselves, which may influence the responses and the analysis? Were participants given the opportunity to review the transcript of their interview for accuracy? Lines 193, 243, 333 contain spelling or grammatical errors

Author Response

We would like to thank the two anonymous reviewers for their support of our research, and their useful feedback.  We have responded to Reviewer 2’s suggestions as below.

Reviewer 2

Comments and Suggestions for Authors

Thank you for undertaking this study to improve understanding of the barriers and enablers those who require haemodialysis as a life sustaining treatment experience relating to food and nutrition. The patient experience and perspective is clearly described and highlights a number of areas where the delivery of nutrition information can be improved. However there is inadequate information in the manuscript about the setting and how dietary information is provided, and methodological considerations, particularly around data collection and the consideration of bias and assumptions during the inductive process of deriving themes.

We have added further information as outlined below.

The manuscript would be improved if more context could be provided around how and when dietary information is provided to this group. For example, is dietary information tailored to individuals, or is generic information provided to everyone?

How often are patients able to access dietitians to ask questions?

Are patients provided with feedback regularly about positive and negative dietary compliance, or are they only approached when potentially non-compliant?

Thank you for this feedback.  We have revised section 3.3.1 in order to address these questions. 

How were the interview questions derived and who was involved?

What was the impetus for the research?

We have clarified this point in the introduction:

A recent New Zealand study undertaken by our team has shown that few patients on haemodialysis are able to meet all of these nutritional requirements[6]

There is inadequate information about the research team included in the manuscript. What relationship exists between the researchers and participants? What bias, assumptions and background do the researchers identify about themselves, which may influence the responses and the analysis? Were participants given the opportunity to review the transcript of their interview for accuracy?

Thank you for this feedback.  We have revised the methods section of the paper in order to clarify these points.

Lines 193, 243, 333 contain spelling or grammatical errors. 

Thank you, we have revised these sections.

Round 2

Reviewer 2 Report

Thank you for making the suggested changes.